# Passive Smoking Is Associated with Multiple Heavy Metal Concentrations among Housewives in Shanxi Province, China

**DOI:** 10.3390/ijerph19148606

**Published:** 2022-07-14

**Authors:** Huiting Chen, Jigen Na, Hang An, Ming Jin, Xiaoqian Jia, Lailai Yan, Nan Li, Zhiwen Li

**Affiliations:** 1Key Laboratory of Reproductive Health, National Health Commission of the People’s Republic of China, Institute of Reproductive and Child Health, School of Public Health, Peking University, Beijing 100191, China; chenhuiting0@pku.edu.cn (H.C.); najigen@bjmu.edu.cn (J.N.); anhang@bjmu.edu.cn (H.A.); 1610306233@bjmu.edu.cn (M.J.); jiaxq@bjmu.edu.cn (X.J.); 2Department of Epidemiology and Biostatistics, School of Public Health, Peking University, Beijing 100191, China; 3Department of Laboratorial Science and Technology, School of Public Health, Peking University, Beijing 100191, China; yll@bjmu.edu.cn

**Keywords:** heavy metals, passive smoking, hair sample, housewives

## Abstract

Background: Passive smoking may increase the content of heavy metals in housewives. However, this association remains a subject of debate. Female passive smoking is widespread, particularly in Chinese rural areas. Objective: This study aimed to assess the association between heavy metal accumulation and passive smoking status among rural housewives. Methods: 405 women were recruited in Shanxi Province of Northern China, and 384 (94.8%, 384/405) participants were included in the final study, of whom 117 women were exposed to passive smoking. The information on their basic characteristics was collected via a structured questionnaire. We used inductively coupled plasma mass spectrometry (ICP-MS) to analyze the concentrations of nine heavy metals, including cadmium (Cd), germanium (Ge), arsenic (As), lead (Pb), titanium (Ti), copper (Cu), iron (Fe), cobalt (Co), and chromium (Cr), in hair samples. Results: The results indicated that higher As, Ge, Ti, and Fe concentrations were significantly associated with passive smoking. After adjusting for potential confounders, the adjusted odds ratios and the 95% confidence intervals of As, Ge, Ti, and Fe were (1.80 (1.13–2.90), *p* = 0.028), (1.78 (1.14–2.80), *p* = 0.007), (1.70 (1.09–2.67), *p* = 0.019), and (1.67 (1.07–2.63), *p* = 0.035), respectively. The statistically significant linear trend of the adjusted odds ratios at different levels further supported their association. Conclusion: Our research concluded that exposure to environmental tobacco smoke might contribute to As, Ge, Ti, and Fe accumulation among housewives.

## 1. Introduction

Tobacco smoke has been a major global public health issue, which could increase the risk of breast cancer, cervical carcinoma, and adverse pregnancy outcomes [1,2]. Passive smoking, which has been commonly observed in multinomial categorical distributions pertaining to smoking status [3], accounts for more than 1.2 million deaths every year globally [4]. Passive smokers inhale not only mainstream smoke (15% of cigarette smoke) exhaled by smokers, but also sidestream smoke (85% of cigarette smoke) that is emitted from the smoldering ends of cigarettes while smokers are not puffing [5]. Sidestream smoke contains numerous cytotoxic substances, especially heavy metals (HMs), in quantities much higher than those found in mainstream cigarette smoke [6]. The active smoking rate of Chinese women was only 2.1% [7], while their passive smoking rate was as high as 68.1% [8]. The majority of Chinese women, particularly those in rural areas, remain unaware of the detrimental health effects of passive smoking. Although previous studies have revealed that tobacco smoke contains more than 6000 kinds of compounds [9], whether heavy metal accumulations are associated with passive smoking is still unclear.

Exposure to heavy metals (HMs) is well-known to have significant impacts on human health [10]. Besides the conventional effects, HMs have also been confirmed to induce reproductive toxicity that ensues in endometriosis, infertility, and abortion among women [11]. Previous studies have reported that active smoking might result in HM accumulation in the human body [12,13]. Emerging evidence has shown that titanium (Ti), arsenic (As), manganese (Mn), and copper (Cu) concentrations in the blood of active smokers were significantly higher than those of nonsmokers [13]. Another study found that arsenic (As), cadmium (Cd), and lead (Pb) levels were significantly higher in the lung tissue of active smokers [12]. However, research about the relationship between passive smoking and HM concentrations is still scant.

Due to the high rate of passive smoking among women in Central and Western China [14], this study aims to examine the association between passive smoking and nine HMs in the hair specimens among rural housewives of Shanxi Province of Northern China, including cadmium (Cd), germanium (Ge), arsenic (As), lead (Pb), titanium (Ti), copper (Cu), iron (Fe), cobalt (Co), and chromium (Cr). Hair sampling is a non-invasive and safe health assessment method. It is used to competently monitor the internal accumulation of HMs as well as blood or tissue specimens [15]. The content in the hair sample can indicate chronic exposure levels to HMs [16].

## 2. Methods

### 2.1. Participants and Recruitment

We carried out a cross-sectional study in Pingding County Hospital in Shanxi Province of Northern China from August 2012 to May 2013, to assess the effects of indoor air pollution on the health of local housewives. The housewives were recruited if they were concordant to the following criteria: (1) resided in Pingding County; (2) had no substantial change in their living situation in the past decade; and (3) were aged 30 or over.

Researchers collected the characteristics of these participants through a questionnaire during face-to-face interviews, mainly including passive smoking status, birth date, occupation, education, and status of a stove for heating. Physical examinations such as height and weight measurements were performed by local physicians according to standard protocol. BMI (body mass index, kg/m^2^) was computed as the value of weight divided by the square of height to be a possible confounding variable. The biomedical ethics committee of Peking University approved our protocol, and all participants signed the consent form.

According to consensus methods, passive smoking in this study was defined as nonsmokers exposed to tobacco smoke at least once a week for at least half an hour, at home or in public places [3]. A total of 405 women were recruited for this study. After excluding 12 active smokers and 9 with missing data about passive smoking, 384 (94.8%, 384/405) participants were included in the final analysis. 

### 2.2. Hair Sample Collection and Laboratory Analysis

Trained professionals cut a strand of hair sample as close as possible to the scalp from the occipital part of the head. These samples were kept in closed, labeled polyethylene zip-lock bags until they were tested in the laboratory. It has been indicated that the average growth of hair is approximately 1 cm per month [17]. We took 24 cm of hair from the roots (weighing approximately 25 mg) to indicate medium-to long-term exposure. Our previously published article elaborated on the details of the laboratory analysis [18]. Briefly, we cut the eligible hair into 1 cm segments. Those samples and the blank vials were sequentially washed with 1 mL of Triton X-100 (Sigma-Aldrich, St. Louis, MO, USA) one time, 1 mL of deionized water three times, and 1 mL of acetone (J.T. Baker^®^, Center Valley, PA, USA) three times. All of the above washes were performed under a vortex for 5 min each time. These samples were then digested with 1 mL of nitric acid in a 15 mL quartz digestion tube in a microwave digester (Ultra WAVE, Milestone, Milan, Italy) for 50 min. Experimenters used inductively coupled plasma mass spectrometry (ICP–MS; ELAN DRC Ⅱ, PerkinElmer, Waltham, MA, USA) to detect the concentrations of HMs. We used the standard hair references (GBW09101a) from the China National Standard Materials Center as a reference standard for quality control. The differences between measured and referred concentrations were mostly <10%, which showed reasonable recovery rates. The standard curve fitting for each metal proved to be satisfactory (all R^2^ > 0.99). Each sample corresponded to three procedural blanks and one reagent blank, and the concentrations were calculated by subtracting the average of the corresponding blanks from the detected concentrations. The HM concentrations were finally converted into values in the unit of ng/g hair. The China Metrology Accreditation (CMA) system qualified our quantitative analysis protocol.

### 2.3. Statistical Analysis

We used the interquartile range (IQR) and median to describe hair HM concentrations due to their skewed distribution. The Mann–Whitney test (nonparametric test) was used to evaluate the difference in HM concentrations between passive smokers and non-passive smokers. The odds ratios (OR) with their 95% confidence intervals (CI) were calculated to estimate the association between HM concentrations and passive smoking. Considering that BMI, age, occupation, and a stove for heating may also affect HM concentrations, they were taken as potential covariates and adjusted in the multiple regression models. We further divided As, Ge, Ti, and Fe concentrations into four levels according to their quartiles, due to their significant effects on passive smoking. We took the lowest level as the control group and calculated each group’s OR to see the dose–response relationship. R software (version 4.0.2; R Development Core Team, Vienna, Austria) was used for all statistical analyses, and a two-tailed *p* value < 0.05 was regarded as statistically significant.

## 3. Results

Table 1 summarizes the sociodemographic characteristics of the 384 participants. The average age and BMI were 52.6 (SD: ± 10.4) years and 24.9 (SD: ± 3.0) kg/m^2^, respectively. Most of the participants were farmers (77.9%), and 74.5% of them used a stove for heating in winter. There were 117 (30.4%) passive smokers among the participants.

Table 2 shows the concentrations of HMs between the passive smoking group and the non-passive smoking group. The detection rates of all the HMs were 100%. The hair HM concentrations were highest for Fe, whose median concentration was 16,405.81 ng/g, followed by Cu (8616.68 ng/g), Ti (4527.54 ng/g), Pb (1342.55 ng/g), As (146.85 ng/g), Cr (120.16 ng/g), Ge (55.38 ng/g), Cd (23.86 ng/g), and Co (12.48 ng/g) (Table 2). A tendency was noted for passive smokers to have higher concentrations for almost all the observed HMs (Cd, Ge, As, Pb, Ti, Cu, Fe). We plotted a bar chart to show the differences in HM concentrations between passive smoking women and non-passive smoking women (Figure 1). The results of the Mann–Whitney test showed that passive smokers had significantly higher concentrations of Ge (59.96 vs. 53.60 ng/g, *p* < 0.01), As (172.28 vs. 131.55 ng/g, *p* < 0.05), Ti (4861.90 vs. 4418.62 ng/g, *p* < 0.05), and Fe (18,331.57 vs. 15,714.06 ng/g, *p* < 0.05) than those in non-passive smokers (Table 2). However, this study did not find a significant association between passive smoking status and Co, Cd, Cr, Pb, and Cu concentrations in the housewives’ hair specimens.

The association between HMs and passive smoking is presented in Table 3. The odds ratio (OR) is a common index in epidemiological research, and a value greater than 1.0 implies that the factor is a risk factor. The results indicated that passive smoking women had significantly higher concentrations of Ge, As, Ti, and Fe (all *p* values < 0.05), and the odds ratios and their corresponding 95% confidence intervals (95% CIs) were Ge: 1.86 (1.20–2.91), As: 1.77 (1.14–2.76), Ti: 1.68 (1.09–2.62), and Fe: 1.68 (1.09–2.62), respectively. Adjusted odds ratios after controlling for potential covariates were further calculated. The adjusted odds ratios still indicated that passive smoking was a factor contributing to the accumulations of Ge, As, Ti, and Fe among housewives. Passive smoking was associated with a 78% increase in the risk of higher levels of Ge (adjusted OR = 1.78, 95% CI: 1.14–2.80), an 80% increase in the risk of higher levels of As (adjusted OR = 1.80, 95% CI: 1.13–2.90), a 70% increase in the risk of higher levels of Ti (adjusted OR = 1.70, 95% CI: 1.09–2.67), and a 67% increase in the risk of higher levels of Fe (adjusted OR = 1.67, 95% CI: 1.07–2.63), respectively. Figure 2 revealed the dose–response relationships between Fe, Ti, As, and Ge levels and passive smoking status. When those four HM levels were divided into four categories, the results of the trend test showed significant linear trends of adjusted ORs on the whole, which indicated that increasing levels of Fe, Ti, As, and Ge had an association with passive smoking (Figure 1, *p* < 0.05).

## 4. Discussion

Female passive smoking in China is widespread and severe due to the high proportion of men smoking, particularly in rural areas [3]. According to a survey conducted by the Chinese Center for Disease Control and Prevention in 2018, the passive smoking rate of Chinese women was up to 68.1% [8]. The relationship between active smoking and higher HM concentrations has been well established. However, whether passive smoking is also associated with higher HM accumulations is controversial. This cross-sectional study indicated that the higher concentrations of As, Ge, Ti, and Fe in hair were significantly associated with passive smoking among housewives. The apparent linear trend of the different levels’ ORs strengthened their association.

In accordance with our results, a Greek study also demonstrated a similar association between passive smoking and a higher arsenic concentration among pregnant women [19]. Arsenic is an endocrine disruptor as defined by the WHO [20]. Arsenic exposure was also found to have a distinct exposure–response relationship with lung cancer [21], indicating that increasing arsenic concentrations might mediate the association between passive smoking and lung cancer. To the best of our knowledge, this is the first time that higher Ge, Ti, and Fe concentrations were found to be significantly associated with passive smoking. Increasing evidence has shown that disorders of cellular Fe metabolism might contribute to a large number of lung diseases, including asthma, cystic fibrosis, chronic obstructive pulmonary disease, and lung cancer [22,23]. Higher levels of iron were also found in both current and former smokers [24].

There are still no studies on the relationship between tobacco smoke and concentrations of Ge and Ti. Studies have shown that mainstream tobacco smoke contains titanium oxide particles [25], which may induce elevated lipid peroxide levels and lung inflammation after being inhaled [26]. Higher Ge or Ti contents caused by exposure to environmental tobacco smoke can potentially impact human health. Ge is an essential trace element in the human body, while excessive Ge can hurt the kidneys, respiratory system, and neural system [27,28,29]. As far as we were concerned, this study was the first to observe the relationship between increasing Ge and Ti concentrations and exposure to tobacco smoke. More studies are recommended in the future to verify the conclusion.

The median concentrations of toxic heavy metals (Pb and Cd) in our study were higher than those in French residents (Pb: 1342.55 vs. 410 ng/g, and Cd: 23.83 vs. 11 ng/g) [30]. The results might be related to the fact that Shanxi Province used to be one of the world’s leading coal-producing areas [31] and had higher environmental exposure to Pb and Cd. Our study also found that Pb and Cd concentrations were higher in passive smokers, but those associations were not statistically significant. The influence of passive smoking on Pb and Cd accumulations is still controversial to date. Three studies reported that passive smoking is related to higher levels of Pb and Cd among children and pregnant women [32,33,34], while others did not find statistically significant associations among children [35,36]. Whether there is a statistically significant correlation between Pb and Cd concentrations with passive smoking might not be the same across different populations.

The research has three limitations in the interpretation of our findings. First, the cross-sectional design cannot provide a direct causal relationship between HM concentrations and passive smoking. Therefore, longitudinal designs may be needed in future studies. Second, passive smoking data were collected using self-reported measures, with no objective biomarker data (e.g., nicotine or cotinine levels) obtained. Therefore, recall and reporting biases may have affected our results. Third, participants in this study were all Han (China’s predominant ethnic group), so our results may not be generalizable to other races. Last, our study investigated the association between passive smoking and multiple heavy metals, and we could not distinguish between conventional cigarettes and special cigarettes, such as waterpipe tobacco or electronic cigarettes. However, alternative tobacco products are rarely used among adults in China, especially among those living in rural areas [37]. Waterpipe tobacco and electronic cigarettes have been identified to cause lung damage in recent years [38]. More research is needed for the subdivided fields to supplement the relevant gaps in the future. Although our research was conducted about a decade ago, the living conditions of the recruited subjects have not changed much over the past 10 years. It hardly affects the authenticity and validity of the association from an epidemiological perspective.

However, our study also had several strengths. To the best of our knowledge, this is the first study focusing on the correlation between nine heavy metals and passive smoking status among normal housewives. The results of our study would provide relevance to the prevention of passive smoking among Chinese housewives. Second, the living conditions of all the participants had no remarkable changes in more than a decade, which minimized the bias caused by other latent significant environmental exposure. Third, the detection rates of all the HMs were 100%, which herein consolidated the concentrations of HMs and their association with passive smoking status. Fourth, hair sampling is a non-invasive technique that is competent in providing objective data for median- and long-term exposure to HMs. Serum and urine are also used as specimens to detect HM concentrations. However, those two specimens could only reflect 24 h or short-term HM exposure [39]. Hair sampling is an ideal method for showing medium- and long-term HM exposure.

## 5. Conclusions

In conclusion, this study demonstrated a significant positive association between As, Ge, Ti, and Fe concentrations and passive smoking among non-smoking women, which provided potential explanations for the harmful mechanism caused by passive smoking. Passive smoking is an everyday exposure for Chinese women; therefore, more measures should be taken to increase awareness of the HMs causing health risks associated with passive smoking.

## Figures and Tables

**Figure 1 ijerph-19-08606-f001:**
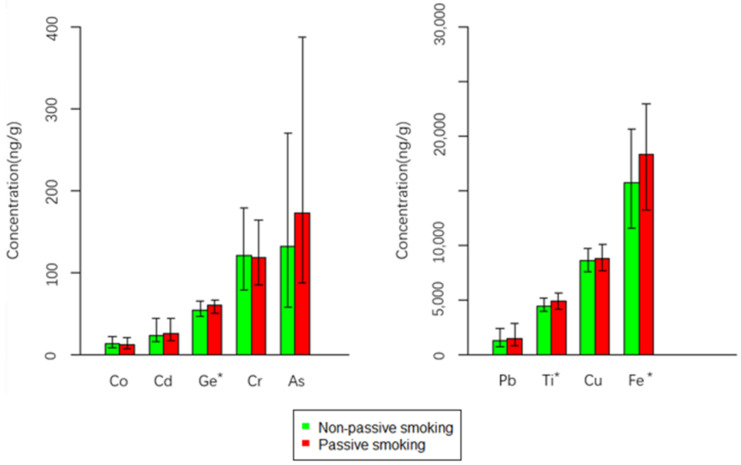
The median and inter-quartile range (IQR) of heavy metal concentrations in passive smokers (*n* = 117) and non-passive smokers (*n* = 267). *: Significantly different at *p* < 0.05, according to Mann–Whitney tests.

**Figure 2 ijerph-19-08606-f002:**
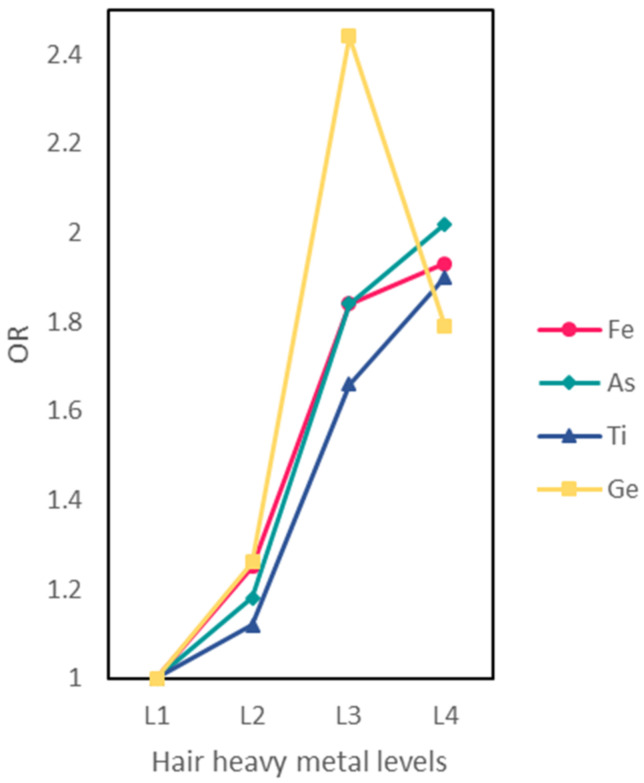
The odds ratio (OR) of heavy metal concentrations and passive smoking. Metal concentrations were classified into four levels by quartiles of all subjects, i.e., <1st quartile (L1), 1st–2nd quartile (L2), 2nd–3rd quartile (L3), >3rd quartile (L4).

**Table 1 ijerph-19-08606-t001:** Basic characteristics of participants in Pingding county, Shanxi Province, Northern China, 2012–2013.

Variable	Number (%)
Age (year, mean ± S.D.)	52.6 ± 10.4
<45	112 (29.2)
45–55	109 (28.4)
55–65	105 (27.3)
≥65	58 (15.1)
Body mass index (kg/m2, mean ± S.D.)	24.9 ± 3.0
<25	214 (55.7)
≥25	170 (44.3)
Occupation	
Non-farmer	85 (22.1)
Farmer	299 (77.9)
Education	
Primary or lower	200 (52.1)
Junior high	113 (29.4)
High school or junior college	38 (9.9)
Above junior college	33 (8.6)
Using a stove for heating	
No	98 (25.5)
Yes	286 (74.5)
Second-hand smoking status	
No	267 (69.5)
Yes	117 (30.5)

S.D.: standard deviation.

**Table 2 ijerph-19-08606-t002:** Hair heavy metal concentrations (ng/g) of passive smoking women and non-passive smoking women.

Metals	P_25_	Median	P_75_	Range	LOD	LOQ	DF (%)	*p* Value ^a^
Co	8.14	12.48	20.74	1.72–27595.60	0.001	0.003	100	
NPS	8.46	12.73	21.45	1.72–469.72				0.271
PS	7.50	12.12	20.15	1.80–27595.60				
Cd	15.97	23.86	43.78	4.85–6546.20	0.003	0.009	100	
NPS	15.41	23.34	43.45	5.04–6546.2				0.432
PS	16.58	25.74	44.03	4.85–1430.95				
Ge	47.52	55.38	66.02	30.80–157.80	0.009	0.027	100	
NPS	46.69	53.60	64.74	30.8–129.63				0.007
PS	49.67	59.96	66.59	40.26–157.8				
Cr	79.34	120.16	172.60	35.38–908.44	0.002	0.006	100	
NPS	79.19	120.47	179.00	36.32–908.44				0.627
PS	85.09	118.09	163.93	35.38–888.6				
As	63.23	146.85	309.54	4.40–3214.49	0.05	0.15	100	
NPS	58.14	131.55	270.79	12.27–3214.49				0.028
PS	87.23	172.28	388.49	4.40–3073.75				
Pb	748.87	1342.55	2433.78	206.92–17,948.57	0.012	0.036	100	
NPS	716.64	1260.75	2351.50	206.92–17,948.57				0.186
PS	789.73	1468.74	2860.55	275.55–15,527.87				
Ti	4012.96	4527.54	5347.90	3038.81–17,023.87	0.01	0.03	100	
NPS	3970.24	4418.62	5171.90	3038.81–15,998.49				0.019
PS	4140.87	4861.90	5660.70	3315.92–17,023.87				
Cu	7628.89	8616.68	9762.22	3812.92–211,613.19	0.03	0.09	100	
NPS	7558.78	8584.14	9741.35	3812.92–211,613.19				0.460
PS	7703.29	8776.28	10,051.64	5156.21–14,8879.4				
Fe	12,095.61	16,405.81	21,933.06	5946.29–98,278.89	0.9	2.7	100	
NPS	11,568.85	15,714.06	20,615.38	5946.29–98,278.89				0.035
PS	13,251.20	18,331.57	22,947.21	6079.57–55,322.95				

DF: detection frequency. LOD: limit of detection. LOQ: limit of quantification. NPS: non-passive smoking. PS: passive smoking. ^a^: *p* values came from Mann–Whitney tests.

**Table 3 ijerph-19-08606-t003:** Odds ratios (ORs) for the risk of higher heavy metal levels associated with passive smoking.

Metals	Crude OR ^a^	Crude OR 95%CI ^a^	Adjusted OR ^b^	Adjusted OR 95%CI ^b^
Co	0.88	0.57–1.37	0.88	0.56–1.36
Cd	1.25	0.81–1.93	1.21	0.78–1.89
Ge *	1.86	1.20–2.91	1.78	1.14–2.80
Cr	0.93	0.60–1.43	0.93	0.59–1.44
As *	1.77	1.14–2.76	1.80	1.13–2.90
Pb	1.38	0.89–2.14	1.31	0.84–2.05
Ti *	1.68	1.09–2.62	1.70	1.09–2.67
Cu	1.08	0.70–1.66	1.05	0.67–1.63
Fe *	1.68	1.09–2.62	1.67	1.07–2.63

^a^: calculated with unconditional binary logistic regression; ^b^: calculated with unconditional binary logistic regression adjusted for BMI, age, occupation, a stove for heating. *: *p* < 0.05.

## Data Availability

The data can be obtained from the corresponding authors upon reasonable request.

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
