# Peer review of "Passive Smoking Is Associated with Multiple Heavy Metal Concentrations among Housewives in Shanxi Province, China"

_ijerph, 2022, doi:10.3390/ijerph19148606_

Round 1
Reviewer 1 Report
This study aimed to assess the association between heavy metal accumulation and passive smoking status among rural housewives. The article is written clearly with a good English. Up-to-date references are included. The statistical analysis is appropriate and the statistical program up to date. The topic is interesting, the study is valuable and topical. The study has some limitations and strengths that are clearly stated in the Discussion. The research concluded that exposure to secondhand smoke might contribute to As, Ge, Ti, and Fe accumulation among housewives. The study demonstrated a significant positive association between As, Ge, Ti, and Fe concentrations and passive smoking among nonsmoking women, which provided potential explanations for the harmful mechanism caused by passive smoking. The authors state that measures should be taken to increase awareness of health risks of heavy metals associated with passive smoking.
Comments and remarks
Passive smoking exposure is a topic of great concern for public health because of its well-known adverse effects on human health. Please check the term secondhand smoke and thirdhand smoke properly.
Please see publications:
Protano, C.; Vitali, M. The New Danger of Thirdhand Smoke: Why Passive Smoking Does Not Stop at Secondhand Smoke. Environ. Health Perspect. 2011, 119, a422.
Matt, G.E.; Quintana, P.J.E.; Destaillats, H.; Gundel, L.A.; Sleiman, M.; Singer, B.C.; Jacob, P.;Benowitz, N.;Winickoff, J.P.; Rehan, V.; et al. Thirdhand tobacco smoke: Emerging evidence and arguments for a multidisciplinary research agenda. Environ Health. Perspect. 2011, 119, 1218–1226.
Other remarks:
1. Table 1, the heading should be more informative
2. Did you ask about the length of marriage in those housewives?
3. How questions about passive smoking were formulated?
4. Did you check only conventional cigarettes?
5. Did you ask about the use of Alternative Tobacco Products (ENDS, Heat-not-Burn Products?
Please, add those remarks into Discussion.
I suggest to accept the article after those Minor revisions.
Reviewer 2 Report
Reviewer
Initial comments
This work is very important, as it brings a lot of information about passive smokers who are at risk and need to be informed.
Abstract:
Comment:
Lines 16-17… Methods: We recruited 384 women in Shanxi Province of northern China,
Lines 78-80 in 2. Methods
2.1. Participants and recruitment
405 women were recruited for this study. After excluding twelve active smokers and nine with missing data about passive smoking, 384 (94.8%, 384/405) participants were included in the final analysis.
Please, correct the methods in the Abstract.
405 were recruited and there are 384 participants….
Line 19…. We used ICP-MS to analyze the concentrations of nine heavy metals,
Please, spell ICP-MS in full.
1. Introduction
Comment:
Lines 49-50…Emerging evidence showed that Ti, As, Mn, and Cu concentrations…..
Please, spell out Ti, As, Mn, and Cu.
Line 51….Another study found that As, Cd, and Pb levels
Please, write Cd, and Pb in full
2. Methods
2.1. Participants and recruitment
Comment:
Line 73……BMI (kg/m2) was computed
Please put BMI in full
2.2. Hair sample collection and laboratory analysis
Comment:
It is suitable
2.3. Statistical analysis
Comment:
It is suitable
3. Results
Comment:
It is suitable
4. Discussion
Comment:
Lines 198-200….The median concentrations of toxic heavy metals (Pb and Cd) in our study were higher than those in French residents (Pb: 1342.55 vs. 410 ng/g, and Cd: 23.83 vs. 11 ng/g) (Goullé et al., 2005).
(Goullé et al., 2005). And the reference?
Lines 230-235….In conclusion, this study demonstrated a significant positive association between As, Ge, Ti, and Fe concentrations and passive smoking among nonsmoking women, which provided potential explanations for the harmful mechanism caused by passive smoking. Passive smoking is an everyday exposure for Chinese women; therefore, more measures should be taken to increase awareness of the HMs-caused health risks associated with passive smoking.
Couldn't the conclusion be separated from the discussion?
5. Conclusion?
Reference
Comment:
Or References?
Lines 198-200….The median concentrations of toxic heavy metals (Pb and Cd) in our study were higher than those in French residents (Pb: 1342.55 vs. 410 ng/g, and Cd: 23.83 vs. 11 ng/g) (Goullé et al., 2005).
(Goullé et al., 2005). ….Missing references!
Thank you
Reviewer 3 Report
The manuscript tilted “Passive smoking is associated with multiple heavy metal con- 2 centrations among housewives in Shanxi Province, China” can not be published in Int. J. Environ. Res. Public Health.
The design of the study is well prepared but the results presented in this paper refer to 10 years ago (“We carried out a cross-sectional study in Pingding County Hospital in Shanxi Province of northern China from August 2012 to May 2013 to assess the effects of indoor air pollution on the health of local housewives”). During last 10 years is possible to change the habits and exposure to tobacco smoke in Chinese women, what makes the results inadequate to real time. With the above in mind, I think that this manuscript should be rejected.
Reviewer 4 Report
In the submitted manuscript “ijerph-1782234” entitled “Passive smoking is associated with multiple heavy metal concentrations among housewives in Shanxi Province, China” Chen et al. aimed to assess the association between heavy metal accumulation and passive smoking status among rural housewives. It is not surprising that passive smoking increase the amount of heavy metals. However the study contain enough novelty for publication. However before acceptance the following questions should be clarified:
1. The higher level of germanium and titanium is interesting. It should be highlight the source of its heavy metals from cigarette.
2. It should also clarify why these elements were investigated. For example Hg could also be interesting.
3. Some additional data regarding analysis are also necessary. Do you use calibration, LOD, LOQ etc.
4. I suggest to check some article from special issue and cite some relevant article
Round 2
Reviewer 3 Report
I keep my attention from the previous review.
